# Evaluation of Dynamic Load Reduction for a Tractor Semi-Trailer Using the Air Suspension System at all Axles of the Semi-Trailer

**Dang Viet Ha [1,2], Vu Van Tan [1,*], Vu Thanh Niem [2] and Olivier Sename [3]**

1 Department of Automotive Mechanical Engineering, Faculty of Mechanical Engineering, University of Transport and Communications, Hanoi 100000, Vietnam; hadv2011@gmail.com

2 Vietnam Register, 18 Pham Hung Street, My Dinh 2 Ward, Nam Tu Liem District, Hanoi 100000, Vietnam; vuthanhniem@gmail.com

3 University Grenoble Alpes, CNRS, Grenoble INP, GIPSA-Lab, 38000 Grenoble, France; olivier.sename@grenoble-inp.fr

\* Correspondence: vvtan@utc.edu.vn; Tel.: +84-947-215-885

**Abstract:** The air suspension system has become more and more popular in heavy vehicles and buses to improve ride comfort and road holding. This paper focuses on the evaluation of the dynamic load reduction at all axles of a semi-trailer with an air suspension system, in comparison with the one using a leaf spring suspension system on variable speed and road types. First, a full vertical dynamic model is proposed for a tractor semi-trailer (full model) with two types of suspension systems (leaf spring and air spring) for three axles at the semi-trailer, while the tractor's axles use leaf spring suspension systems. The air suspension systems are built based on the GENSYS model; meanwhile, the remaining structural parameters are considered equally. The full model has been validated by experimental results, and closely follows the dynamical characteristics of the real tractor semi-trailer, with the percent error of the highest value being 6.23% and Pearson correlation coefficient being higher than 0.8, corresponding to different speeds. The survey results showed that the semi-trailer with the air suspension system can reduce the dynamic load of the entire field of speed from 20 to 100 km/h, given random road types from A to F according to the ISO 8608:2016 standard. The dynamic load coefficient (*DLC*) with the semi-trailer using the air spring suspension system can be reduced on average from 14.8% to 29.3%, in comparison with the semi-trailer using the leaf spring suspension system.

**Keywords:** tractor semi-trailer; air suspension system; air spring; leaf spring; dynamic load reduction; rad safety

## 1. Introduction

### 1.1. Background

Tractor semi-trailers are an important means of freight transport worldwide. In the US, about 11 billion tons of commodities are transported every year by this mode, valued at USD 700.4 billion, creating jobs for approximately 10 million people. According to statistics, there were about 5.6 million tractor semi-trailers [1] in the US in 2014. The number of tractor semi-trailers in various European countries as of 2012 was as follows: approximately 310,000 vehicles in France, nearly 290,000 in Germany, more than 270,000 in Poland, more than 250,000 in Spain, more than 200,000 in Turkey, more than 130,000 in The Netherlands, approximately 100,000 in Italy, close to 80,000 in Romania, and approximately 50,000 in The Czech Republic [2]. Tractor semi-trailers have made remarkable contributions to the socio-economic growth and transportation sector development, yet due to the nature of the heavy load of these vehicles, roads need protecting in a serious manner. The key factor to road damages is the road surface pressure on roads in contact with these vehicles. There are two fundamental solutions to reduce the vertical dynamic tire forces on roads, i.e.,

(1) to increase the contact surface area of vehicle wheels and/or (2) to reduce the vertical dynamic tire forces in contact with the road (including static load and dynamic load). The static load of vehicles is subject to each national regulation, and hence the desired load reduction for vehicle wheels during operation will target the reduction of dynamic load. A study by Cebon [3] has introduced solutions for reduced pressure on roads, including: (1) use of balanced vehicle axles; (2) double tires; (3) static load sharing, and (4) reduction of dynamic load. Solutions of balanced vehicle axles, double tires, and static load sharing are used for increasing the road contact surface. Heavy vehicles have all applied these solutions. Hence, besides the above-mentioned solutions, it is also necessary to reduce the dynamic load of tractor semi-trailers, which can be realized by the two most commonly considered options, specifically: controlled suspension systems and non-linear elastic element suspension systems, such as air springs. The controlled suspension system is not cost-effective for this means of transport because of the high production costs and high energy consumption of the control system. An optimal, feasible option of the air suspension system can be substituted for the conventional leaf spring suspension system. In order to form the basis for evaluating the extent of dynamic load reduction in the air suspension system against the leaf spring suspension systems, it is important to establish a full vertical dynamic model of tractor semi-trailers using both of the suspension systems mentioned above, to identify the characteristics of the elastic element, given that the other structural parameters of a vehicle remain constant. A common criteria used to compare the dynamic tire force is the dynamic load coefficient (*DLC*). Dynamic load is impacted by two key drivers, movement velocity and road types, according to each static load of vehicles.

*1.2. Related Works*

1.2.1. Air Suspension Model

What makes the air suspension system different from the leaf spring suspension systems is the air spring. The air spring was first successfully manufactured in the 1950s when flexible rubber fiber came into being. In the early days, the air spring was used in buses in the US and in Europe. After that, manufacturers started to design air springs for heavy vehicles with leveling valves for pressure adjustment. This application was then introduced in cars in the later years. The air spring in heavy vehicles, in the US, gradually replaced the conventional leaf spring due to the benefits that the air suspension system brought about. Until 1996, the air suspension system accounted for 36% of heavy vehicles' suspension systems, and then 75% in 2008 [4]. Then, specialized studies looked further into the air suspension system with a focus on the following issues: (1) creating and developing the models of the air suspension system, and (2) integrating such models of the air suspension system to vehicle models for dynamic vehicle studies.

In terms of structure, the air suspension system is made of the following key parts: the air spring, reservoir, pipe, and leveling valve. In order to establish the model of the air suspension system, the literature shows the modeling of thermodynamic processes in the system, corrections of structural parameters, and descriptions of loss in the system. Currently, popular models of the air suspension system include: Nishimura, Vampire, Simpac, GENSYS, Quaglia, and Cebo, and relevant studies have tackled the modeling of the air suspension system, as detailed below.

Development of the Model of the Air Suspension System

Wang [5] developed the model of the air suspension system using Quaglia's methodology [6] considering thermal exchange, based on the conservation of energy principle, the identification of dynamic stiffness of the air spring via experiments, and the development of the characteristics of the leveling valve with the correction coefficient identified via experiments. The research was implemented based on a quarter car model. Similar to the air suspension system model based on the Quaglia method, Chen [7] established a model of the air suspension system with a full description of the following compartments: the air spring, reservoir, pipe, leveling valve, and the connector, and the model considers thermal

exchange and system loss. The research resulted in the identification of two important input parameters, i.e., effective area (as a function of displacement) and the characteristic of leveling valves. The work was performed with a quarter vehicle model. The authors integrated the air suspension system model to the tractors to study the balance of the suspension system, along with validation experiments. However, the research did not look into the dynamic load. Zhu et al. [8] established the dynamic model of the air spring according to the Quaglia model, which considered friction and viscosity. The research focused on 6 input parameters of the model, including effective area and the volume of the air spring as the spare root function of displacement, friction, and viscosity, via experiments. The research was carried out on a quarter model with the air suspension system to improve the accuracy of the system.

Nieto et al. [9] established an analytical model of the air suspension system based on experimental descriptions. The non-linear model described three components: the air spring, reservoir, and pipe, without a description of the leveling valves. The research resulted in the identification of effective area and volume of the air spring according to the displacement of the air spring. However, the research only established a quarter model, with a focus on the convoluted bellow. Chang and Lu [10] introduced a dynamic model of the air spring with consideration of the thermal transmission process. The model itself is a complex of three blocks: the "geometry block", describing the effective area and the volume of the air spring according to the displacement of the air spring, the "thermodynamics block", describing the state of thermodynamics in the system, and the "calculate spring force block", identifying the forces of the air spring, as calculated. The survey results from a quarter model with the sine excitation function were compared between the classical model and the new model. The research also performed validation experiments in a quarter model and integrated the model of the air suspension system in the vehicle, using co-simulation. However, the air suspension system model integrated on passenger vehicles was limited to automotive models, without further surveys. Chen et al. [11] established a non-linear model for a semi-trailer with multiple axles, combined with the longitudinal connected air suspension, for the purpose of evaluating the dynamic load sharing using the *DLC* and the dynamic load sharing coefficient (DLSC). The research mainly focused on evaluating the impacts of the structural factors (pressure, pipe diameters, connector diameters) on *DLC* and DLSC. However, this research did not evaluate the impacts of velocity and road types. White [12] developed a model of the air suspension system and looked into the impacts of different structures of the leveling valve on the rollover stability of heavy vehicles. The research results are related to leveling valves and their correlation with rollover limits. Nakajima et al. [13] developed a model of the air suspension system, which described the air mass flow rate as a function of the valve opening angle. The valve opening angle is dependent on the displacement of the rotating control arm, as a result of the relative movements between the vehicle frame and the axle. The model was developed for rail transport.

Integrating the Model of the Air Suspension System in Vehicle Models

Several researchers have studied the application of the GENSYS and Quaglia models to understand the suspension systems (structure and control), specifically the cab suspension system. The following paragraphs outline studies that integrated the suspension system model into the vehicle model.

Sayyaadi and Shokouchi [14] used the GENSYS model to study the impacts of the structural parameters of a suspension system in a vehicle. The research developed a horizontal and vertical quarter model, considered friction, and identified dynamic stiffness in relation to force-displacement and the dynamic stiffness in the frequency zone. It also studied the impacts of structural parameters, such as the volume of the reservoir and the diameters and lengths of pipes to calculate the design of the suspension system in the vehicle. Moheyeldein et al. [15] established two quarter vehicle models: the classical model and the GENSYS model, using the random excitation function, and studied the

impacts of suspension system structural parameters such as ride height, diameters of the air spring, pressure of the air spring, volume of the reservoir, and diameter and length of the pipes. The authors compared the classical model with the GENSYS model by means of acceleration of the sprung mass and the dynamic load, thereby confirming the preciseness of the GENSYS model. Abid [16] conducted research establishing an air suspension system model equivalent to the passive suspension system model of passenger automobiles. In the research, a quarter model using the GENSYS model and a quarter model of the passive suspension system were studied, with the aim to find out the optimal specifications for the air suspension system according to relative displacement, similar to a passive suspension system given the same input excitation. This was achieved through using the OptiY programming for minimizing the variations when comparing the displacements of the two models.

Tang [17] established the air spring model to suspend the cab in accordance with the Quaglia model, whereas the parameters of effective area and the volume of the air spring can be identified via experiments. Co-simulation between ADAMS and AMESim was used to integrate the air spring model in the dynamic model of automobile cabs. Comparisons were made between the results of experiments and simulations and established according to the ride comfort under random road excitation. The purpose of the study was to understand the ride comfort by means of acceleration at the position of the driver in the frequency range; however, the study did not examine dynamic loads. Hondo [18] established a model and experiment to identify the relation between the input pressure of leveling valves and the pressure of the air spring by means of the air mass flow rate through the valve under the impact of centrifugal inertia force.

Studies of active and semi-active control for the air suspension system include research by Razdan et al. [19], who recommended a responsive control system and active fluctuation control through the internal air flow rate control in the system, and introduced the algorithms for controlling the air mass flow rate through valves according to different parameters. This research used a quarter model and the air suspension system model for control calculations.

In general, the studies showed that the air suspension system model is established based on the descriptions of the dynamics and thermodynamics of the air flow in the system. The complexity of the model depends on the specialty and preciseness of the structural parameters, and the calculations of losses, so that different models can be formulated. The model of the air suspension system can be used to study the dynamics of wheels, to optimize the suspension system design, or to establish vehicle models.

1.2.2. Dynamic Load

Buhari et al. [20] studied the *DLC* of heavy vehicles. They established a quarter model using the air suspension system with different structures: single axles, double axles, and triple axles, of various suspension systems (air spring and leaf spring), as the basis for comparison. The parameters used in the study include: load level, with random excitation on type C roads, at the speed of 30 m/s. The research results brought forward some important conclusions, as follows: (1) dynamic load is influenced by four variables: vehicle velocity, road type, load level, and the type of suspension system; (2) the load level 1/3 of the full load causes the most severe damage to the road, compared with other load levels, and (3) the *DLC* of vehicles with the air suspension system is always smaller than that using leaf spring suspension systems. The research used a quarter model and conditions of a common road type and fixed velocity. Siddiqui [21] conducted an analysis of the impacts of dynamic loads of urban buses on roads and the influential factors on dynamic load, and studied ride comfort of urban buses in many seat locations in the passenger compartment. Study results focused mainly on the relation between dynamic loads and the ride comfort of urban buses. Muluka [22] established a dynamic model for 3-axle trucks using the air suspension system with consideration of the air spring, factors impacting the dynamic load of vehicle wheels, and optimizing the anti-shock function, as well as the

elasticity unit for less dynamic load. It is in this research where a vertical half-model 3-axle truck was established. Hu [23] pointed out the variables affecting the *DLC* of a heavy-duty truck, including: excitation frequency, velocity, and load level. The research established the relation between the *DLC* and a function of velocity versus the road with a high roughness value and versus the road with a low roughness value. Research results were generated on a quarter model of trucks with two road types and three load levels. However, the scope of the study was still limited, and a full-function automobile model has not been considered yet.

In the literature, several studies have been concerned with various vehicles, ranging from trucks, passenger automobiles, to tractor semi-trailers, with relevant content about factors affecting dynamic load and comparisons of dynamic load of suspension systems and optimization of design parameters of suspension systems for reduced dynamic load. The variables affecting the dynamic load have been investigated at a general level. The above-mentioned studies suggest that the use of the air suspension system according to the GENSYS model and the application of a full vertical dynamic model for tractor semi-trailers are new and require more strategic evaluation in terms of impacting factors of dynamic loads, which include vehicle velocity and road types.

*1.3. Paper Contributions*

The purpose of this paper is to evaluate the possibility to reduce the dynamic load of a semi-trailer by using the air suspension system against the leaf spring suspension system. The contributions of this paper are listed as follows:

- Propose a full vertical dynamic model of a 6-axle tractor semi-trailer with 26 degrees of freedom for the leaf spring suspension systems and 28 degrees of freedom for the air suspension systems. Leaf suspension systems are used for the tractor and the air suspension system for the semi-trailer. These two suspension systems are presented as two sub-models, in which the air suspension system is based on the GENSYS model and integrated into the vehicle model.
- Validate the full model through real experiments using measurement parameters, including: displacement of un-sprung mass at the 4th axle, relative displacement of sprung and un-sprung mass at the 4th axle (suspension space), acceleration of sprung mass at the 4th axle, and movement velocity of the vehicle. The validation results between the research model and the experiments at different velocities showed that the Pearson correlation coefficient is higher than 0.8, and the percent error of the highest value is smaller than 6.3%.
- Comparisons of the *DLC* on the 4th, 5th, and 6th axles of the semi-trailer with the two types of suspension system in diversified driving conditions, including: velocity range from 20 to 100 km/h and random types of roads according to ISO 8608:2016, ranging from road A to road F. Comparative results will show that the air suspension system could, on average, reduce dynamic load by up to 29.3% compared to the conventional semi-trailer using the leaf spring suspension systems.

## 2. Vehicle Modeling of a Tractor Semi-Trailer

*2.1. Full Model of a Tractor Semi-Trailer*

A full vertical dynamic model of a tractor semi-trailer (full model) is developed in Figure 1, where the tractor comprises of three axles using the leaf spring system, where the 2nd and 3rd axles are balance axles. The semi-trailer has three axles with two types of suspension systems (leaf spring and air spring), as two sub-models. The full model with leaf spring suspension contains 26 degrees of freedom, whereas the one with the air suspension system has 28 degrees of freedom. Air spring is considered as a high-pressure compressed air reservoir with variable volume. The compression and expansion of the air spring not only changes the volume but also the thermodynamic state of the air inside of the air spring. Therefore, the air spring stiffness is nonlinear; meanwhile, the leaf spring

stiffness can be considered with a constant value (described in Sections 2.2.1 and 2.2.2). Tire stiffness is also considered with a constant value (in Equation (A13) of Appendix A).

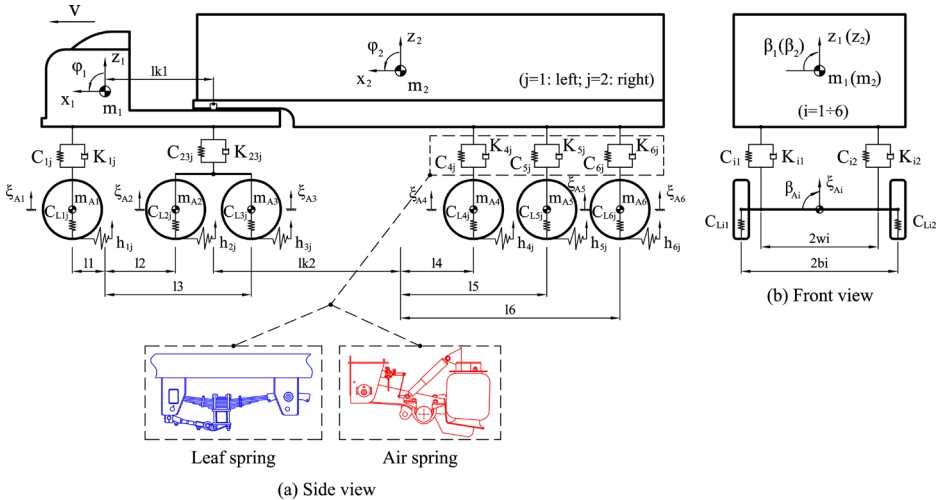

**Figure 1.** Full vertical dynamic model of a tractor semi-trailer using two types of suspension systems: (**a**) Side view, (**b**) Front view.

The full model was established based on the multi-body system (MBS) method applying the Newton–Euler Equations [24]. The equations for the tractor semi-trailer are defined as follows:

$$m_1(\ddot{z}_1 - \dot{x}_1\dot{\varphi}_1) = (F_{C11} + F_{K11} + F_{C12} + F_{K12}) + (F_{C231} + F_{K231} + F_{C232} + F_{K232}) - F_{kz1} \tag{1}$$

$$J_{y1}\ddot{\varphi}_1 = \sum_{j=1}^{2} \left( \begin{array}{c} -l_1(F_{C1j} + F_{K1j}) + (\frac{l_2+l_3}{2})(F_{C23j} + F_{K23j}) \\ -(h_1 - r_{11})(F'_{x1j} + F'_{x2j} + F'_{x3j}) - (M_{1j} + M_{2j} + M_{3j}) \end{array} \right) \\ -(h_{w1} - h_1)F_{wx1} + (h_1 - h_{k1})F_{kx1} - l_{k1}F_{kz1} \tag{2}$$

$$J_{x1}\ddot{\beta}_1 = w_1(F_{C11} + F_{K11} - F_{C12} - F_{K12}) + w_2(F_{C231} + F_{K231} - F_{C232} - F_{K232}) \\ -M_{T1} - M_{T2} - M_{T3} \tag{3}$$

$$m_2(\ddot{z}_2 - \dot{x}_2\dot{\varphi}_2) = \sum_{j=1}^{2}(F_{C4j} + F_{K4j} + F_{C5j} + F_{K5j} + F_{C6j} + F_{K6j}) + F_{kz2} \tag{4}$$

$$J_{y2}\ddot{\varphi}_2 = \sum_{j=1}^{2} \left( \begin{array}{c} l_4(F_{C4j} + F_{K4j}) + l_5(F_{C5j} + F_{K5j}) + l_6(F_{C6j} + F_{K6j}) \\ -(h_1 - r_{41})(F'_{x4j} + F'_{x5j} + F'_{x6j}) - (M_{4j} + M_{5j} + M_{6j}) \end{array} \right) \\ -(h_{w2} - h_2)F_{wx2} - (h_2 - h_{k2})F_{kx2} - l_{k2}F_{kz2} \tag{5}$$

$$J_{x2}\ddot{\beta}_2 = \sum_{i=4}^{6}(w_i(F_{Ci1} + F_{Ki1} - F_{Ci2} - F_{Ki2}) - M_{Ti}) \tag{6}$$

$$\begin{cases} m_{A1}\ddot{\xi}_{A1} = (F_{CL11} + F_{CL12}) - (F_{C11} + F_{K11} + F_{C12} + F_{K12}) \\ J_{Ax1}\ddot{\beta}_{A1} = b_1(F_{CL11} - F_{CL12}) + w_1(F_{C12} + F_{K12} - F_{C11} - F_{K11}) + M_{T1} \end{cases} \tag{7}$$

$$\begin{cases} m_{Ai}\ddot{\xi}_{Ai} = (F_{CLi1} + F_{CLi2}) - (F_{CKi1} + F_{CKi2}) \\ J_{Axi}\ddot{\beta}_{Ai} = b_i(F_{CLi1} - F_{CLi2}) + w_i(F_{CKi2} - F_{CKi1}) + M_{Ti} \end{cases} \quad (\text{with } i = 2;3) \tag{8}$$

$$\begin{cases} m_{Ai}\ddot{\xi}_{Ai} = (F_{CLi1} + F_{CLi2}) - (F_{Ci1} + F_{Ki1} + F_{Ci2} + F_{Ki2}) \\ J_{Axi}\ddot{\beta}_{Ai} = b_i(F_{CLi1} - F_{CLi2}) + w_i(F_{Ci2} + F_{Ki2} - F_{Ci1} - F_{Ki1}) + M_{Ti} \end{cases} \quad (\text{with } i = 4;5;6) \tag{9}$$

In the Equations (2), (4), and (5), j = 1 for the left wheels and j = 2 for the right wheels. The equations determine the binding force, as specified in Appendix A. The symbols and values of the parameters of the model are presented in Appendix B.

### 2.2. Suspension Models of Semi-Trailer

2.2.1. Leaf Spring Model

The leaf spring suspension system for the semi-trailer is a continuously balanced suspension system. The sets of leaf springs on each axle are connected to each other through the longitudinal balance bar (equalizer), as shown in Figure 2. It is this binding mode that creates a dynamic balance of each axle. Hence, it is necessary to express the dynamic equation for this equalizer. The binding force, $F_{Cij}$, of the suspension system while the stiffness of the leaf spring remains constant is:

$$F_{Cij} = C_{ij}(\xi_{Dij} - z_{uij}) \tag{10}$$

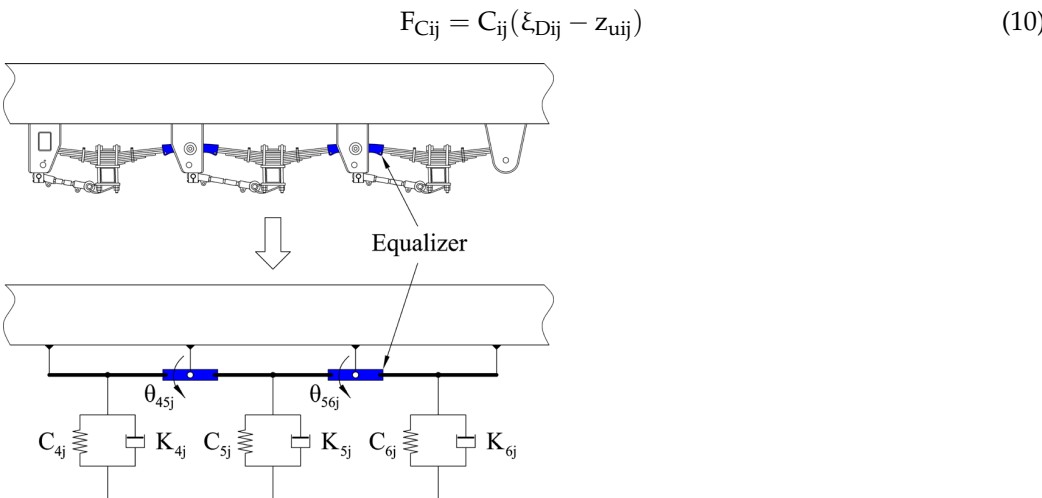

**Figure 2.** Suspension model of the semi-trailer using the leaf spring.

The dynamic equations of the equalizer in the suspension system are as follows:

$$J_{y45j}\ddot{\theta}_{45j} = \frac{c}{4}(F_{C5j} + F_{K5j} - F_{C4j} - F_{K4j}) \tag{11}$$

$$J_{y56j}\ddot{\theta}_{56j} = \frac{c}{4}(F_{C6j} + F_{K6j} - F_{C5j} - F_{K5j}) \tag{12}$$

In Equations (10)–(12), i = 1 ÷ 6, j = 1 for the left wheels, and *j* = 2 for the right wheels.

2.2.2. Air Suspension Model

This sections presents the model of the air suspension system based on the deployment of the GENSYS model [25,26], as shown in Figure 3. The structure of the air suspension system semi-trailer comprises of three components: the air spring, the reservoir, and the pipes (Figure 3a). It is modeled according to the GENSYS model with three components: the elastic component ($C_{ez}$, $C_{vz}$), the viscous nonlinear damping component ($K_{z\beta}$), and the mass of the circulating air flow (M), as shown in Figure 3b. In this study, the air suspension model with the air spring element using the GENSYS model was combined with the vehicle model, and therefore the following assumptions were used: ignore the internal friction of the system, and the effect of hysteresis is due to the internal friction.

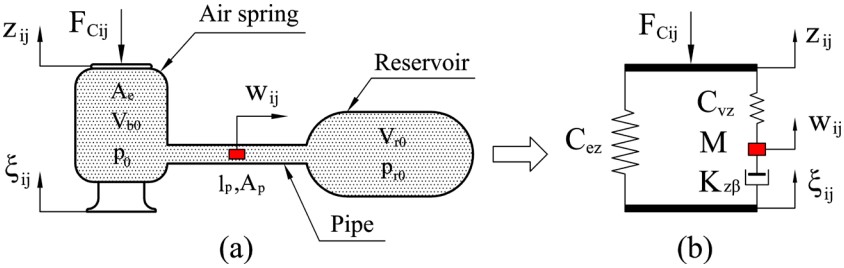

**Figure 3.** Model of the air suspension system: (**a**) Physical model, (**b**) GENSYS model.

Displacements $z_{ij}$, $\xi_{ij}$, and $w_{ij}$ are the displacements of the upper part of the air spring, the lower part of the air spring, and of the air flow inside the pipes, respectively.

The static load is the function of the air in the air spring as follows:

$$F_{zt} = (p_0 - p_a)A_e \tag{13}$$

Equations to determine the binding force of the air suspension system are as follows:

$$F_{Cij} = C_{ez}(\xi_{ij} - z_{ij}) + C_{vz}(\xi_{ij} - z_{ij} - w_{ij}) \tag{14}$$

$$M\ddot{w}_{ij} = C_{vz}(\xi_{ij} - z_{ij} - w_{ij}) - K_{z\beta}\left|\dot{w}_{ij}\right|^2 \text{sign}(\dot{w}_{ij}) \tag{15}$$

$$C_{ez} = \frac{p_0 A_e^2 n}{V_{b0} + V_{r0}} \tag{16}$$

$$C_{vz} = C_{ez}\frac{V_{r0}}{V_{b0}} \tag{17}$$

$$M = l_p A_p \rho \left(\frac{A_e}{A_p}\frac{V_{r0}}{V_{b0} + V_{r0}}\right)^2 \tag{18}$$

$$K_{z\beta} = K_s\left(\frac{A_e}{A_p}\frac{V_{r0}}{V_{b0} + V_{r0}}\right)^3 \tag{19}$$

$$K_s = \frac{1}{2}\rho k_t A_p \tag{20}$$

### 2.3. Random Road Profile

Random road profiles are formulated according to the ISO standard 8608:2016 [27]. By using the sinusoidal method, the height, $h(x)$, of the road profile is determined in the following equation:

$$h(x) = \sum_{i=1}^{N}\sqrt{2G_d(n_i)\Delta n}\cos(2\pi i\Delta nx + \varphi_i); \; \Delta n = \frac{1}{L} \tag{21}$$

where $\varphi_i$ is the random phase taken from $[0 \ldots 2\pi]$ (rad), and L is the length of the road section created randomly (m). The random road profiles according to the ISO standard 8608:2016, formulated from Equation (21), are shown in Figure 4.

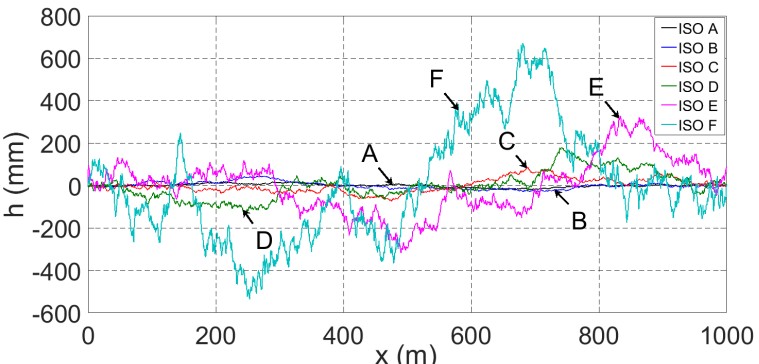

**Figure 4.** The road profiles according to the ISO standard 8608:2016 (A–F).

## 3. Evaluation Criteria

The dynamic load of the semi-trailer at each vehicle axle is assessed by the dynamic load coefficient (*DLC*) in Formula (22) [28]. The greater the value of *DLC*, the higher the

dynamic load: for movement safety and road protection, this value is desired to be as small as possible.

$$DLC_i = \frac{\text{RMS}(F_{CL(i)})}{F_{z(ti)}} \tag{22}$$

$$\text{RMS}(F_{CL(i)}) = \sqrt{\frac{1}{T}\int_0^T F^2_{CL(i)} dt} \tag{23}$$

Here, $F_{CL(i)}$ is the dynamic load of the wheels on axle i (N), $F_{z(ti)}$ is the static load of the wheels on axle i in the vertical direction (N), and T is the time of measurement/study (s).

In order to evaluate the extent of the reduction of the dynamic load of a semi-trailer at each axle, Formula (24) is used to compare the efficiency of the air suspension system with that of the leaf spring suspension system:

$$\Delta DLC_i(\%) = \frac{DLC_i^{Leaf} - DLC_i^{Air}}{DLC_i^{Leaf}} \times 100 \tag{24}$$

where $DLC_i^{Leaf}$, $DLC_i^{Air}$ are the values of the *DLC* in the models of leaf spring and air suspension systems, in respective axles. In Formulas (22)–(24), *i* = 4, 5, and 6.

## 4. Validation of the Tractor Semi-Trailer Model Using Real Experiments

### 4.1. Experimental Purpose

The experimental purpose was to validate the model by comparing the test parameters and the simulated ones. The experiments were conducted to measure the parameters of the vehicle in the vertical direction, given the bump cosine excitation, at different speeds. The vehicle used in the experiments was a semi-trailer branded DOOSUNG DV-CSKS-400AR-1, linked with a tractor branded HYUNDAI HD700, as shown in Figure 5. The tractor has three axles, with the two rear axles using the leaf suspension system, and the semi-trailer has three axles using the air suspension system.

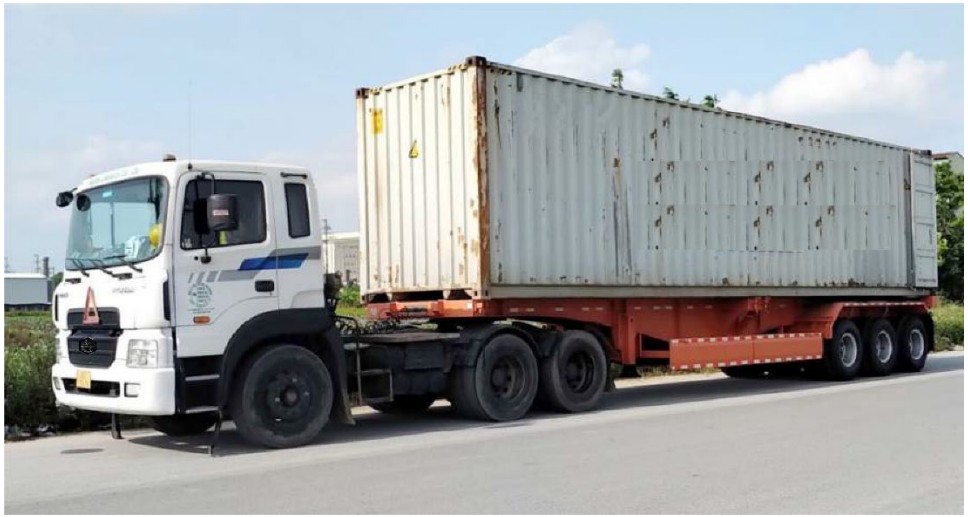

**Figure 5.** Tractor semi-trailer used in the experiment.

### 4.2. Equipment for the Experiment

The measurement equipment, comprising of 4 sensors, a signal-processing device, and a computer, connected as shown in Figure 6, specifically included:

- Displacement measurement sensor (02 units), HF-750C by Kistler (Ostfildern, Germany).
- Acceleration measurement sensor (01 unit), DYTRAN-3263A2 by DYTRAN Instruments, Inc. (Chatsworth, CA, USA).

- Vehicle speed sensor (01 chiếc), Correvit S-motion DTI model 2055A by Kistler (Ostfildern, Germany).
- Signal processor set, model SIRIUS, a set of 3 types: SIRIUSi-8xSTGM+, SIRIUSi-8xACC, and SIRIUSi-8xCAN, by DEWEsoft (Trbovlje, Slovenia).
- A computer for control and displaying measurement results.

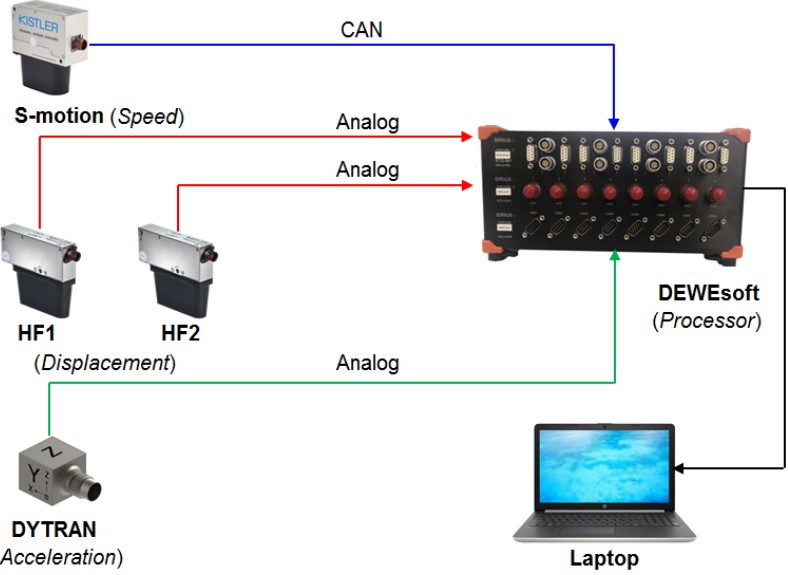

**Figure 6.** Connection diagram of experimental equipment.

Sensors were installed on the vehicle at the location of the 4th axle (on the vehicle axle and frame) and the rear position, as in Figure 7. The HF1 sensor was placed on the vehicle frame for measuring relative displacements between the vehicle frame and the axle. The HF2 sensor was placed underneath the axle for measuring axle displacement, the DYTRAN sensor was placed on the vehicle frame for measuring the acceleration of the vehicle body, and the S-motion sensor was placed at the end of the vehicle for measuring the speed of the vehicle. The S-motion sensor is used to send signals in CAN format, and the remaining sensors are used to send analog signals. Signals from sensors are sent via the processor DEWEsoft for display on the computer screen in real-time mode, allowing visual real-time results.

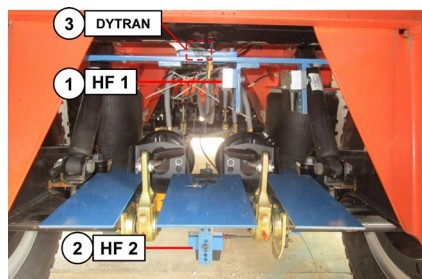 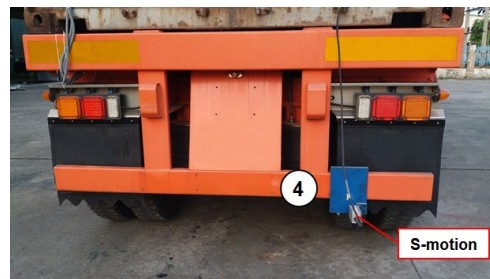

**Figure 7.** Sensor locations.

### 4.3. The Design of Experiments

A bump cosine unit of 50 mm height was used to stimulate the wheels of the two sides. The vehicle moves steadily at speeds changing in the range of 10, 20, and 30 km/h, as described in Figure 8, and the necessary parameters are summarized in Table 1. The measurement results from the moment of maintained constant speed, until the vehicle passes the bump cosine unit. The results are measured instantaneously and processed via the signal processor.

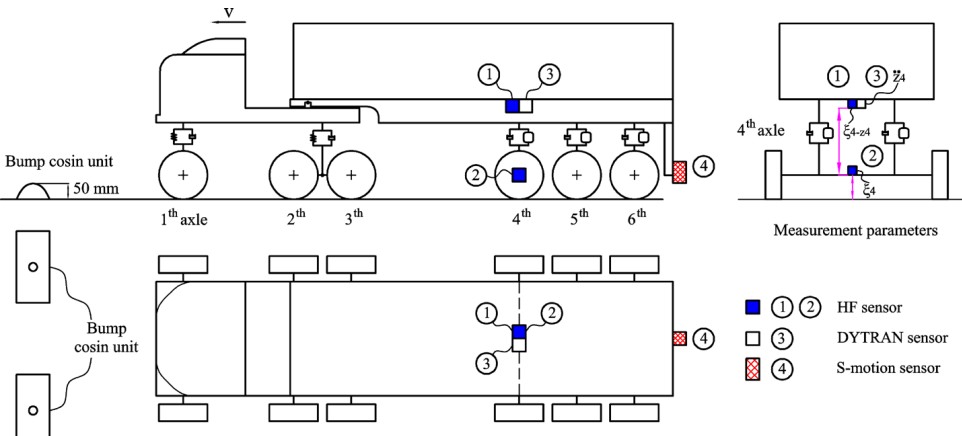

**Figure 8.** Experiment scenario description.

**Table 1.** Necessary parameters.

| No. | Parameters | Symbols | Sensors | Position |
|---|---|---|---|---|
| 1 | Relative displacement of the sprung mass and un-sprung mass at the 4th axle | $\xi_4$-$z_4$ | HF1 | 1 |
| 2 | Displacement of the un-sprung mass at the 4th axle | $\xi_4$ | HF2 | 2 |
| 3 | Acceleration of the sprung mass relative to the 4th axle | $\ddot{z}_4$ | DYTRAN | 3 |
| 4 | Speed of the vehicle | v | S-motion | 4 |

*4.4. Results*

The experimental results corresponding to the speeds of 10, 20, and 30 km/h are displayed in Figures 9 and 10. The parameters displayed include: axle vertical displacement and acceleration of the vehicle body at the point of the 4th axle.

*4.5. Comparison of the Results in Simulation and Experiments*

The correlation coefficient (r) is a statistics ratio that measures the covariance between two variables. The correlation coefficient value ranges from −1 to 1. A value of the correlation coefficient of 0 (or near 0) implies that there is no linear dependency between the two variables; in contrast, correlations equal to −1 or 1 imply that the two variables are absolutely correlated. The value of the correlation coefficient below zero (r < 0) implies that when "x" increases, then "y" decreases (and vice versa, when "x" decreases then "y" increases). If the value of the correlation coefficient is positive (r > 0), it means "x" increases, which can cause "y" to increase, and if "x" decreases, then "y" decreases, accordingly. There are many correlation coefficients, however the most popular is the Pearson correlation coefficient [29], which is defined as follows:

$$r = \frac{\sum\limits_{i=1}^{n}(x_i - \bar{x})(y_i - \bar{y})}{\sqrt{\left[\sum\limits_{i=1}^{n}(x_i - \bar{x})^2\right]\left[\sum\limits_{i=1}^{n}(y_i - \bar{y})^2\right]}} \tag{25}$$

where $x_i$ is the individual spot measurements in the experiment, $\bar{x}$ is the average value of all measurement points in the experiment, $y_i$ is the value at each point of the simulation, and $\bar{y}$ is the average value of all points of the simulation.

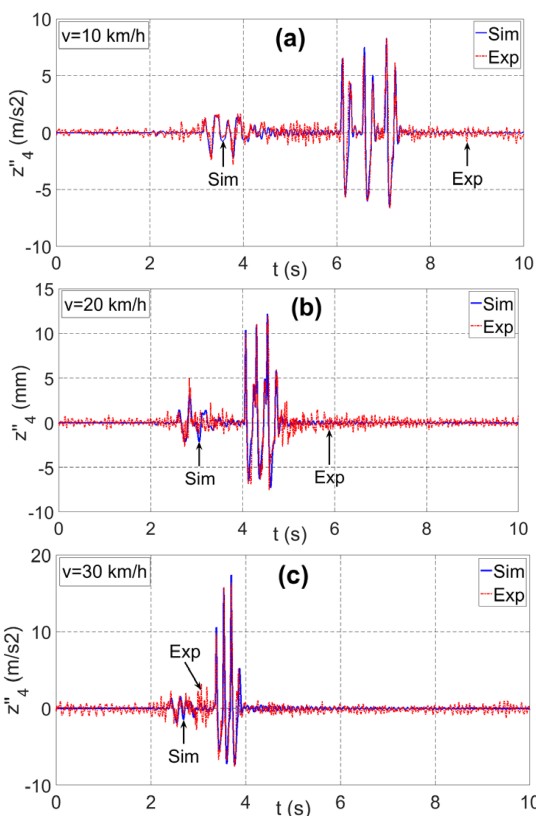

**Figure 9.** Acceleration of the vehicle body at the 4th axle ($z''_4$): (**a**) at 10 km/h, (**b**) at 20 km/h, (**c**) at 30 km/h.

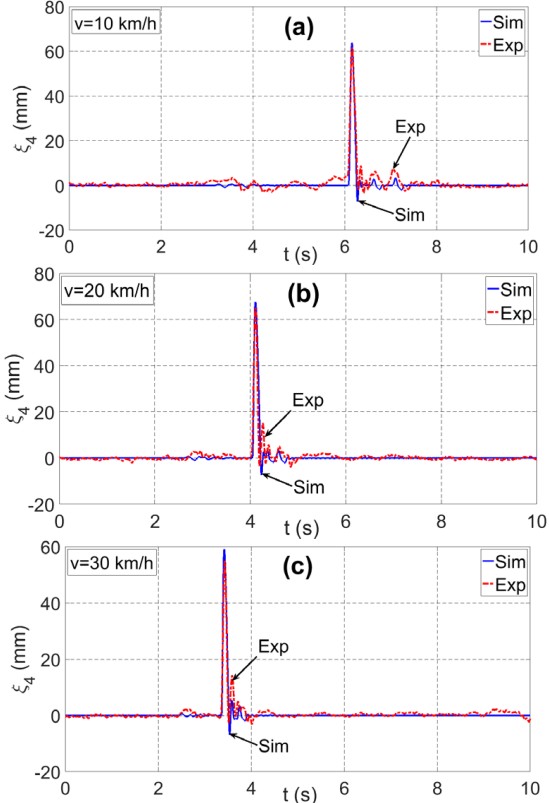

**Figure 10.** Displacement of the 4th axle ($\xi_4$): (**a**) at 10 km/h, (**b**) at 20 km/h, (**c**) at 30 km/h.

The error of the highest value is defined as in Equation (26):

$$\Delta_i(\%) = \frac{y_{i,max} - x_{i,max}}{x_{i,max}} \times 100 \tag{26}$$

where $x_{i,max}$ is the highest value of the experiment and $y_{i,max}$ is the highest value of the simulation.

The comparison results of the simulation and the experiment by Pearson correlation coefficient and the error of the highest value are demonstrated in Tables 2 and 3.

**Table 2.** Pearson correlation coefficient between the simulation and the experiment results.

| Speed | $r_{(\xi4-z4)}$ [-] | $r_{(\xi4)}$ [-] | $r_{(\ddot{z}_4)}$ [-] |
|---|---|---|---|
| 10 km/h | 0.8058 | 0.9449 | 0.8547 |
| 20 km/h | 0.8540 | 0.9192 | 0.8560 |
| 30 km/h | 0.8994 | 0.9530 | 0.8381 |

**Table 3.** Error of the highest value between the simulation and the experiment results.

| Speed | max($\xi_4$–$z_4$) (%) | max($\xi_4$) (%) | max($\ddot{z}_4$) (%) |
|---|---|---|---|
| 10 km/h | 1.80 | 3.45 | 0.24 |
| 20 km/h | 3.45 | 4.12 | 1.35 |
| 30 km/h | 5.89 | 6.23 | 5.78 |

The errors between simulation and experiment results are shown through the Pearson correlation coefficient and the error of the highest value. Tables 2 and 3 show that the error of the highest value is 6.23%, and the correlation coefficients are all greater than 0.8, which proves the preciseness of the vertical dynamic model. The parameters are listed in Appendix B, and closely follow the measured parameters in the experiment.

## 5. Simulation Results Analysis

This study examined the random road types, according to the ISO standard 8608:2016, corresponding to A, B, C, D, E, and F. The tractor semi-trailer's motions correspond to the changing speed, from 20 to 100 km/h, at the speed interval of 10 km/h. The evaluation results are shown in *DLC* values.

Dynamic load ($F_{CL}$) at the 4th, 5th, and 6th axles corresponds to road type A and a speed of 20 km/h, as shown in the time domain in Figure 11. The plots show a smaller dynamic load on all 3 axles of the semi-trailer using the air suspension system than that of the semi-trailer using the leaf spring suspension system.

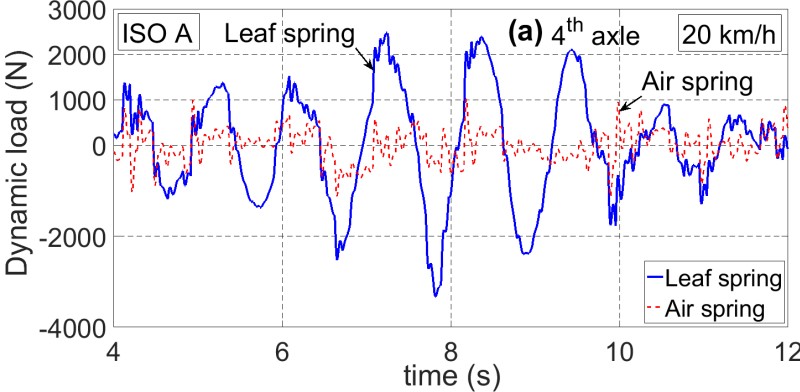

**Figure 11.** *Cont*.

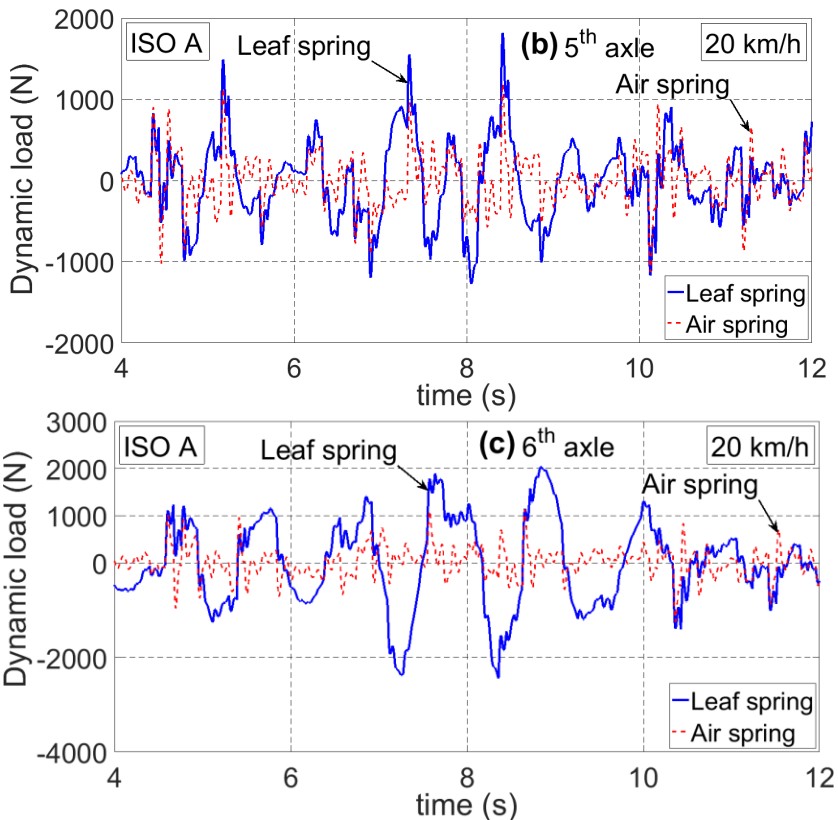

**Figure 11.** Responses to time domain of the dynamic load: (**a**) at the 4th axle, (**b**) at the 5th axle, (**c**) at the 6th axle.

The comparison of *DLC* values on the 4th, 5th, and 6th axles of the semi-trailer using both types of suspension systems, responding to each random road type and speed of vehicles from 20 to 100 km/h, can be seen in Figure 12. The semi-trailer in this study is fully loaded.

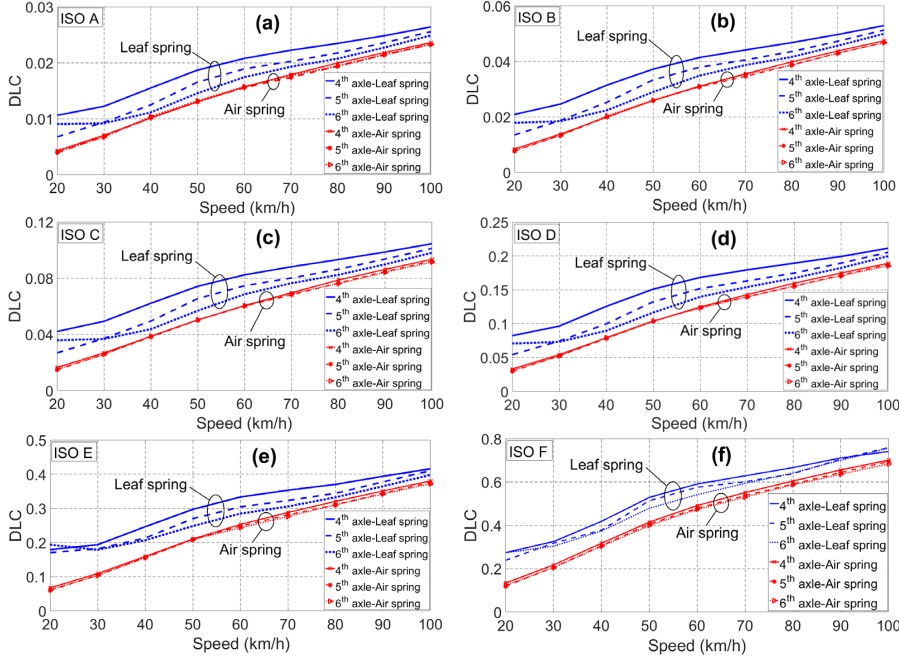

**Figure 12.** Comparison of *DLC* values respective to the two types of suspension systems: (**a**) ISO A, (**b**) ISO B, (**c**) ISO C, (**d**) ISO D, (**e**) ISO E, (**f**) ISO F.

The results in Figure 12 show that when the road type changes from A to F, the value of *DLC* on the axles is increased in both cases of vehicles with leaf spring and air suspension systems. However, the comparison of the two types of suspension systems shows that the value of the *DLC* with the air suspension system is always smaller than that of the *DLC* with the leaf spring suspension system, on all types of roads and all three axles. The decrease in *DLC* value was the lowest on road type A and the highest on road type E, which are shown in the values of $\Delta DLC_i$ (%) in Tables 4–6.

At the 4th axle, the extent of the reduction of the *DLC* ($\Delta DLC4$) was from 5.4% to 62.4%, and at the 5th and 6th axles was in the ranges of 7.4–64.9% and 5.5–68.1%, respectively. On the other hand, the extent to which the *DLC* value changed among the axles of the semi-trailer with the air suspension system is not remarkable (Figure 12), and this is also another advantage of the air suspension system, in terms of sharing of loads among axles.

**Table 4.** The extent of the reduction of $\Delta DLC$ (%): 4th axle.

| $\Delta DLC_4$ (%) | | Random Road | | | | | |
|---|---|---|---|---|---|---|---|
| | | A | B | C | D | E | F |
| Speed (km/h) | 20 | 59.5 | 59.5 | 61.0 | 59.8 | 62.4 | 51.2 |
| | 30 | 42.2 | 43.8 | 45.6 | 43.7 | 43.5 | 33.2 |
| | 40 | 34.6 | 36.5 | 38.0 | 37.1 | 35.3 | 24.1 |
| | 50 | 30.5 | 30.6 | 32.3 | 31.4 | 29.2 | 21.6 |
| | 60 | 24.6 | 25.1 | 26.6 | 25.9 | 24.1 | 16.9 |
| | 70 | 19.5 | 19.5 | 21.0 | 20.4 | 18.5 | 12.4 |
| | 80 | 15.0 | 14.8 | 15.7 | 15.6 | 13.4 | 9.3 |
| | 90 | 12.2 | 12.0 | 12.5 | 12.2 | 10.9 | 7.8 |
| | 100 | 10.6 | 10.3 | 10.6 | 10.5 | 8.6 | 5.4 |
| Average | | 27.6 | 28.0 | 29.3 | 28.5 | 27.3 | 20.2 |

**Table 5.** The extent of the reduction of $\Delta DLC$ (%): 5th axle.

| $\Delta DLC_5$ (%) | | Random Road | | | | | |
|---|---|---|---|---|---|---|---|
| | | A | B | C | D | E | F |
| Speed (km/h) | 20 | 41.9 | 43.1 | 45.3 | 45.1 | 64.9 | 50.1 |
| | 30 | 26.5 | 28.2 | 30.1 | 29.4 | 42.4 | 35.2 |
| | 40 | 18.7 | 21.2 | 22.9 | 21.8 | 27.1 | 19.2 |
| | 50 | 21.1 | 22.3 | 23.4 | 21.8 | 22.5 | 20.9 |
| | 60 | 17.7 | 18.6 | 19.4 | 18.2 | 18.5 | 17.0 |
| | 70 | 14.2 | 14.7 | 15.1 | 14.3 | 13.0 | 10.6 |
| | 80 | 10.9 | 11.0 | 11.4 | 10.9 | 9.7 | 7.4 |
| | 90 | 9.2 | 9.1 | 9.6 | 9.3 | 8.8 | 8.8 |
| | 100 | 8.8 | 8.7 | 8.8 | 8.9 | 8.3 | 8.5 |
| Average | | 18.8 | 19.7 | 20.6 | 20.0 | 23.9 | 19.8 |

**Table 6.** The extent of the reduction of $\Delta DLC$ (%): 6th axle.

| $\Delta DLC_6$ (%) | | Random Road | | | | | |
|---|---|---|---|---|---|---|---|
| | | **A** | **B** | **C** | **D** | **E** | **F** |
| Speed (km/h) | 20 | 54.9 | 55.6 | 57.5 | 56.4 | 68.1 | 54.8 |
| | 30 | 26.4 | 27.9 | 30.1 | 28.7 | 42.1 | 32.5 |
| | 40 | 6.7 | 9.1 | 11.1 | 11.2 | 24.3 | 18.9 |
| | 50 | 9.3 | 9.8 | 11.3 | 10.4 | 16.0 | 16.7 |
| | 60 | 9.7 | 10.8 | 12.1 | 11.9 | 14.6 | 12.6 |
| | 70 | 8.1 | 8.8 | 9.6 | 9.5 | 9.7 | 10.7 |
| | 80 | 6.3 | 7.1 | 7.7 | 7.2 | 6.7 | 8.6 |
| | 90 | 5.5 | 5.9 | 6.3 | 6.4 | 6.3 | 9.3 |
| | 100 | 6.4 | 6.5 | 6.9 | 7.1 | 6.5 | 9.6 |
| Average | | 14.8 | 15.7 | 17.0 | 16.5 | 21.6 | 19.3 |

The *DLC* values of both types of suspension system are shown in Tables 4–6 and are summarized as follows:

— $\Delta DLC$ in both suspension systems became smaller when the vehicle speed was higher. At low speed, the extent of the reduction in *DLC* was remarkable: by 62.4% at the 4th axle, at 20 km/h, on the road type E, by 64.9% at the 5th axle on the road type E, and by 68.1% at the 6th axle on the road type E. When vehicle speed increased, the extent to which *DLC* reduced was smaller: by 5.4% at the speed of 100 km/h, on the road type F, at the 4th axle, by 7.4% at the speed of 80 km/hm on the road type F, at the 5th axle, and by 5.5% at the speed of 90 km/h, on the road type A, at the 6th axle.

— The average $\Delta DLC$ value with both suspension systems gradually reduced from the 4th axle to the 6th axle; at the 4th axle, it was reduced the most, by 29.3%, at the 5th axle by 23.9%, and at the 6th axle by 21.6%.

— At different speeds, the average $\Delta DLC$ values dropped in the range from 14.8% to 29.3%. The axles of the semi-trailer show behavioral differences in Tables 4–6 due to the following reasons:

— The excitation of each axle is not simultaneous, there is a phase difference between axles.

— The geometrical position of each axle relative to the vehicle's center of gravity is different, leading to a different vertical displacement at each axle.

For a clearer understanding of the reduction extent of *DLC* values of the semi-trailer using the air suspension system, 3D graphics was used to demonstrate $\Delta DLC$ (%) of individual axles by speed and road surface, as shown in Figure 13.

The 3D graph in Figure 13 shows that the $\Delta DLC$ value of the semi-trailer using air suspension was lower than that of the semi-trailer using the leaf suspension system. It is shown within the speed range of 20 to 100 km/h, on the road types from A to F, at all three axles (4th, 5th, and 6th) of the semi-trailer. The maximum reductions were seen on road type E at the speed of 20 km/h (62.4% at the 4th axle, 64.9% at the 5th axle, and 68.1% at the 6th axle). The minimum reductions were 5.4% at the 4th axle on road type F at the speed of 100 km/h, 7.4% at the 5th axle on the road type F at the speed of 80 km/h, and 5.5% at the 6th axle on the road type A at the speed of 90 km/h. This shows that the largest reduction occurred on the road type E and at a low speed (20 km/h), while the smallest reduction occurred at a high speed ($\geq$80 km/h).

The survey, evaluation, and experimental results have shown the effectiveness of the air suspension system in reducing the dynamic load of the tractor semi-trailer vehicle. The GENSYS air spring model used in conjunction with the vehicle model was clearly demonstrated in this study. Further studies of the effect of internal friction and the calculation of equivalent mechanical impedance should be considered in assessing the ride

comfort and safety road holding of automobiles in general and of tractor semi-trailers in particular [30,31].

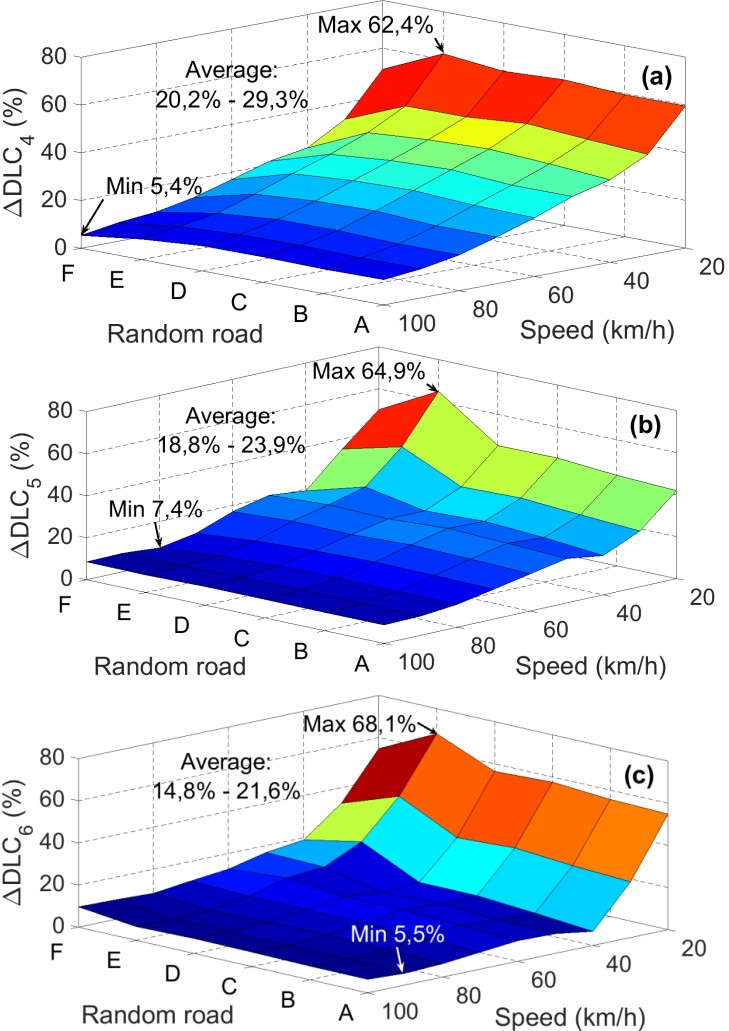

**Figure 13.** The relationship between the extent of $\Delta DLC$ reduction, velocity, and road surface at the 4th, 5th, and 6th axles of the semi-trailer using the air suspension system: (**a**) $\Delta DLC_4$, (**b**) $\Delta DLC_5$, (**c**) $\Delta DLC_6$.

## 6. Conclusions

This paper presented a full vertical dynamic model of a tractor semi-trailer. This model considers a semi-trailer using two types of suspension systems (leaf spring and air spring) and a tractor using leaf spring suspension systems. The air suspension system was modeled on the basis of the GENSYS model. The dynamic model of the semi-trailer has been validated via experiments, with the percent error of the highest value being 6.23% and the Pearson correlation coefficient being higher than 0.8, corresponding to different speeds, thereby confirming the correctness of the research model in comparison with the experimental vehicle. Evaluation and comparative results have shown that the tractor semi-trailer using the air suspension system can reduce the dynamic load across a speed range of 20 to 100 km/h, on all random road types from A to F, in accordance with ISO 8608:2016. *DLC* values can reduce, on average, by 14.8–29.3% compared with the tractor semi-trailer using the leaf spring suspension system. The research results have reinforced the advantages of the air suspension system in heavy vehicles for the purpose of less road damages.

Perspectives of future research can include the development of the models of the tractor semi-trailer using the air suspension system, with the Quaglia model, in which leveling valve attributes are identified via experiments.

**Author Contributions:** Conceptualization, D.V.H., V.V.T., V.T.N. and O.S.; methodology, D.V.H., V.V.T., V.T.N. and O.S.; software, D.V.H., V.V.T. and V.T.N.; validation, D.V.H., V.V.T. and V.T.N.; formal analysis, D.V.H., V.V.T. and V.T.N.; investigation, D.V.H., V.V.T., V.T.N. and O.S.; resources, D.V.H., V.V.T., V.T.N. and O.S.; data curation, D.V.H., V.V.T., and V.T.N.; writing—original draft preparation, D.V.H., V.V.T. and V.T.N.; writing—review and editing, O.S.; visualization, D.V.H. and V.V.T.; supervision, O.S. All authors have read and agreed to the published version of the manuscript.

**Funding:** This research received no external funding.

**Conflicts of Interest:** The authors declare no conflict of interest.

## Appendix A

*Equations to Determine Binding Forces*

Binding forces of a suspension system at the 1st, 2nd, and 3rd axles of the tractor:

$$
\begin{cases}
F_{C1j} = \begin{cases}
C_\infty\left(\xi_{1j} - z_{1j} + f^n_{d1j}\right) & \text{if} \quad f^n_{d1j} < (\xi_{1j} - z_{1j}) \\
C_{1j}\left(\xi_{1j} - z_{1j}\right) & \text{if} \quad f^t_{d1j} \le (\xi_{1j} - z_{1j}) \le f^n_{d1j} \\
-C_\infty\left(\xi_{1j} - z_{1j} - f^t_{d1j}\right) & \text{if} \quad (\xi_{1j} - z_{1j}) < f^t_{d1j}
\end{cases} \\
F_{K1j} = K_{1j}\left(\dot{\xi}_{1j} - \dot{z}_{1j}\right)
\end{cases} ; (j = 1, 2) \quad (A1)
$$

$$
\begin{cases}
F_{C23j} = \begin{cases}
C_\infty\left(\xi_{23j} - z_{23j} + f^n_{d23j}\right) & \text{if} \quad f^n_{d23j} < (\xi_{23} - z_{23j}) \\
C_{23j}\left(\xi_{23j} - z_{23j}\right) & \text{if} \quad f^t_{d23j} \le (\xi_{23j} - z_{23j}) \le f^n_{d23j} \\
-C_\infty\left(\xi_{23j} - z_{23j} - f^t_{d23j}\right) & \text{if} \quad (\xi_{23j} - z_{23j}) < f^t_{d23j}
\end{cases} \\
F_{K23j} = K_{23j}\left(\dot{\xi}_{23j} - \dot{z}_{23j}\right)
\end{cases} ; (j = 1, 2) \quad (A2)
$$

Equations used to determine the points of displacement above or below the suspension system at the 1st, 2nd, and 3rd axles of the tractor:

$$
\begin{cases}
z_{11} = z_1 - l_1 \sin\varphi_1 + w_1 \sin\beta_1 \\
z_{12} = z_1 - l_1 \sin\varphi_1 - w_1 \sin\beta_1 \\
\xi_{11} = \xi_{A1} + w_1 \sin\beta_{A1} \\
\xi_{12} = \xi_{A1} - w_1 \sin\beta_{A1}
\end{cases} \quad (A3)
$$

$$
\begin{cases}
z_{231} = z_1 + \frac{l_2 + l_3}{2}\sin\varphi_1 + \frac{w_2 + w_3}{2}\sin\beta_1 \\
z_{232} = z_1 + \frac{l_2 + l_3}{2}\sin\varphi_1 - \frac{w_2 + w_3}{2}\sin\beta_1 \\
\xi_{231} = \frac{1}{2}(\xi_{A2} + \xi_{A3} + w_2 \sin\beta_{A2} + w_3 \sin\beta_{A3}) \\
\xi_{232} = \frac{1}{2}(\xi_{A2} + \xi_{A3} - w_2 \sin\beta_{A2} - w_3 \sin\beta_{A3})
\end{cases} \quad (A4)
$$

Binding forces of balance suspension systems at the 2nd and 3rd axles of the tractor:

$$
\begin{cases}
F_{CK21} = \frac{1}{2}(F_{C231} + F_{K231}) - \frac{J_{yCB11}\ddot{\theta}_{11}}{a} \\
F_{CK31} = \frac{1}{2}(F_{C231} + F_{K231}) + \frac{J_{yCB11}\ddot{\theta}_{11}}{a}
\end{cases} \quad (A5)
$$

$$
\begin{cases}
F_{CK22} = \frac{1}{2}(F_{C232} + F_{K232}) - \frac{J_{yCB12}\ddot{\theta}_{12}}{a} \\
F_{CK32} = \frac{1}{2}(F_{C232} + F_{K232}) + \frac{J_{yCB12}\ddot{\theta}_{12}}{a}
\end{cases} \quad (A6)
$$

Damping force of suspension systems of the tractor semi-trailer:

$$
F_{Kij} = K_{ij}\left(\dot{\xi}_{ij} - \dot{z}_{ij}\right); (i = 4, 5, 6; j = 1, 2) \quad (A7)
$$

Displacement at the center point of the wheel of the tractor semi-trailer:

$$\xi_{wij} = \xi_{Ai} + (-1)^{j-1} b_i \sin \beta_{Ai} \quad (i = 1 \div 6; j = 1, 2) \tag{A8}$$

Displacement below the suspension system of the semi-trailer:

$$\xi_{ij} = \xi_{Ai} + (-1)^{j-1} w_i \sin \beta_{Ai} \quad (i = 4, 5, 6; j = 1, 2) \tag{A9}$$

Displacement above the leaf spring suspension system of the semi-trailer:

$$\begin{cases} z_{41} = z_2 + l_4 \sin \varphi_2 + w_4 \sin \beta_2 - \frac{c}{4} \sin \theta_{451} \\ z_{42} = z_2 + l_4 \sin \varphi_2 - w_4 \sin \beta_2 - \frac{c}{4} \sin \theta_{452} \\ z_{51} = z_2 + l_5 \sin \varphi_2 + w_5 \sin \beta_2 + \frac{c}{4} \sin \theta_{451} - \frac{c}{4} \sin \theta_{561} \\ z_{52} = z_2 + l_5 \sin \varphi_2 - w_5 \sin \beta_2 + \frac{c}{4} \sin \theta_{452} - \frac{c}{4} \sin \theta_{562} \\ z_{61} = z_2 + l_6 \sin \varphi_2 + w_6 \sin \beta_2 + \frac{c}{4} \sin \theta_{561} \\ z_{62} = z_2 + l_6 \sin \varphi_2 - w_6 \sin \beta_2 + \frac{c}{4} \sin \theta_{562} \end{cases} \tag{A10}$$

Displacement of the upper point in the air suspension system of the semi-trailer:

$$\begin{cases} z_{41} = z_2 + l_4 \sin \varphi_2 + w_4 \sin \beta_2 \\ z_{42} = z_2 + l_4 \sin \varphi_2 - w_4 \sin \beta_2 \\ z_{51} = z_2 + l_5 \sin \varphi_2 + w_5 \sin \beta_2 \\ z_{52} = z_2 + l_5 \sin \varphi_2 - w_5 \sin \beta_2 \\ z_{61} = z_2 + l_6 \sin \varphi_2 + w_6 \sin \beta_2 \\ z_{62} = z_2 + l_6 \sin \varphi_2 - w_6 \sin \beta_2 \end{cases} \tag{A11}$$

Determination of the binding force from the axles to the vehicle body:

$$\begin{aligned} F'_{x11} + F'_{x12} &= f(F_{zt11} + F_{zt12}) \\ F'_{x21} + F'_{x22} + F'_{x31} + F'_{x32} &= f(F_{zt21} + F_{zt22} + F_{zt31} + F_{zt32}) - \frac{M_{21} + M_{22} + M_{31} + M_{32}}{r_{bx}} \\ F'_{x41} + F'_{x42} &= f(F_{zt41} + F_{zt42}) \\ F'_{x51} + F'_{x52} &= f(F_{zt51} + F_{zt52}) \\ F'_{x61} + F'_{x62} &= f(F_{zt61} + F_{zt62}) \end{aligned} \tag{A12}$$

Binding forces between the wheels and the road surface:

$$\begin{aligned} F_{CLij} &= \begin{cases} C_{Lij}(h_{ij} - \xi_{wij}) & \text{if} \quad h_{ij} - (\xi_{wij} - f^t_{ij}) \geq 0 \\ 0 & \text{if} \quad h_{ij} - (\xi_{wij} - f^t_{ij}) < 0 \end{cases} \\ F_{zij} &= F_{CLij} + F_{ztij} \qquad (i = 1 : 1 : 6; j = 1, 2) \end{aligned} \tag{A13}$$

Aerodynamic force:

$$F_{wx1} = C_x A_{x1} \frac{\rho v_0^2}{2} \tag{A14}$$

$$F_{wx2} = C_x A_{x2} \frac{\rho v_0^2}{2} \tag{A15}$$

Active moment:

$$M_{21} = M_{22} = M_{31} = M_{32} = \frac{1}{4}\left((G_1 + G_2)gf + \frac{1}{2}\rho C_x A_{x1} v_0^2\right) r_{21} \tag{A16}$$

## Appendix B

**Table A1.** Vehicle parameters.

| No. | Parameters | Symbol | Value | Unit |
|-----|-----------|--------|-------|------|
| 1 | Distance from CG of tractor to 1st axle | $l_1$ | 1.522 | m |
| 2 | Distance from CG of tractor to 2nd axle | $l_2$ | 1.528 | m |
| 3 | Distance from CG of tractor to 3rd axle | $l_3$ | 2.828 | m |
| 4 | Distance from CG of tractor to 5th wheel | $l_{k1}$ | 1.918 | m |
| 5 | Distance from CG of semi-trailer to kingpin | $l_{k2}$ | 5.377 | m |
| 6 | Distance from CG of semi-trailer to 4th axle | $l_4$ | 2.493 | m |
| 7 | Distance from CG of semi-trailer to 5th axle | $l_5$ | 3.803 | m |
| 8 | Distance from CG of semi-trailer to 6th axle | $l_6$ | 5.113 | m |
| 9 | Tire trace on 1st axle of tractor | $2b_1$ | 2.06 | m |
| 10 | Tire trace on 2nd and 3rd axles of tractor | $2b_2, 2b_3$ | 1.85 | m |
| 11 | Tire trace on 4th, 5th, and 6th axles of semi-trailer | $2b_4, 2b_5, 2b_6$ | 1.84 | m |
| 12 | Width between 2 leaf springs of 1st axle of tractor | $2w_1$ | 0.88 | m |
| 13 | Width between 2 leaf springs of 2nd and 3rd axles of tractor | $2w_2, 2w_3$ | 1.02 | m |
| 14 | Width between 2 air springs of 4th, 5th, and 6th axles of semi-trailer | $2w_4, 2w_5, 2w_6$ | 0.89 | m |
| 15 | Static tire diameter of tractor and semi-trailer | $r_{ij}$ | 0.542 | m |
| 16 | Sprung mass of tractor | $m_1$ | 6490 | kg |
| 17 | Sprung mass of semi-trailer | $m_2$ | 36,210 | kg |
| 18 | Un-sprung mass distributed on 1st axle of tractor | $m_{A1}$ | 570 | kg |
| 19 | Un-sprung mass distributed on 2nd and 3rd axles of tractor | $m_{A2}, m_{A3}$ | 785 | kg |
| 20 | Un-sprung mass distributed on 4th, 5th, and 6th axles of semi-trailer | $m_{A4}, m_{A5}, m_{A6}$ | 750 | kg |
| 21 | Leaf spring stiffness of wheels 11 and 12 of tractor | $C_{11}, C_{12}$ | 250,000 | N/m |
| 22 | Leaf spring stiffness of wheels 231 and 232 of tractor | $C_{231}, C_{232}$ | 1,400,000 | N/m |
| 23 | Tire stiffness of wheels 11 and 12 of tractors | $C_{L11}, C_{L12}$ | 980,000 | N/m |

**Table A1.** *Cont.*

| No. | Parameters | Symbol | Value | Unit |
|---|---|---|---|---|
| 24 | Leaf spring stiffness of wheels 41, 42, 51, 52, 61, and 62 of semi-trailer | $C_{4j}$, $C_{5j}$, $C_{6j}$ | 313,700 | N/m |
| 25 | Tire stiffness of wheels 21, 22, 31, and 32 of tractor | $C_{L2j}$, $C_{L3j}$ | 1,960,000 | N/m |
| 26 | Tire stiffness of wheels 41, 42, 51, 52, 61, and 62 of semi-trailer | $C_{L4j}$, $C_{L5j}$, $C_{L6j}$ | 1,960,000 | N/m |
| 27 | Damping coefficient of wheels 11 and 12 of tractor | $K_{11}$, $K_{12}$ | 15,000 | Ns/m |
| 28 | Damping coefficient of wheels 231 and 232 of tractor | $K_{231}$, $K_{232}$ | 30,000 | Ns/m |
| 29 | Damping coefficient of wheels 41, 42, 51, 52, 61, and 62 of semi-trailer | $K_{4j}$, $K_{5j}$, $K_{6j}$ | 15,000 | Ns/m |
| 30 | Effective area of the air spring (calculated for 1 air spring) | $A_e$ | 0.0483 | $m^2$ |
| 31 | Diameter of the air spring | $D_b$ | 0.248 | m |
| 32 | The inner diameter of the pipe | $d_P$ | 0.01 | m |
| 33 | Pipe length | $l_P$ | 3 | m |
| 34 | Initial pressure of the air spring in full load | $p_0$ | 818,000 | Pa |
| 35 | Pressure of reservoir | $p_r$ | $8.5 \times 10^5$ | Pa |
| 36 | Atmospheric pressure | $p_a$ | $1.0 \times 10^5$ | Pa |
| 37 | Initial volume of the air spring | $V_{b0}$ | $12 \times 10^{-3}$ | $m^3$ |
| 38 | Volume of reservoir (one piece) | $V_{r0}$ | 0.048 | $m^3$ |
| 39 | Specific mass density of atmospheric air in standard condition | $\rho$ | 1.185 | $kg/m^3$ |
| 40 | Adiabatic coefficient | $n$ | 1.4 | / |
| 41 | Non-linear exponential factor | $\beta$ | 1.8 | / |
| 42 | Loss coefficient | $k_t$ | 3.5 | / |

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
