# Peer review of "Evaluation of Dynamic Load Reduction for a Tractor Semi-Trailer Using the Air Suspension System at all Axles of the Semi-Trailer"

_actuators, doi:10.3390/act11010012_

Round 1

Reviewer 1 Report

In this paper a full vertical dynamic model of a tractor semi-trailer (full model) has been set up with 2 types of suspension systems (leaf spring and air spring) for 3 axles at the semi-trailer, while the tractor’s axles use leaf spring suspension systems. After the experimental validation of the vehicle model the dynamical characteristics of the real tractor semi-trailer have been investigated with both types of suspensions.

In section 2.1 please comment on the existing non-linearities of the leaf spring suspensions. Furthermore, what would be the effect of using Pacejka tire model instead of a spring?

What is the diameter of the simulated air spring?

In section 4.5, Table 2 please add the units. Moreover in Tables 2 & 3 you should change «,» το «.».

In section 5, elaborate more on the results of Tables 4 - 6. Why is the behavior of every axle different?

Finally, it would be interesting to provide the percentage of the technical ability of each axle for leaf springs and air spring suspensions in each speed and road profile.

Author Response

Dear Reviewer,
We would like to thank the International Journal of Actuators Editor and Reviewers for the interest they showed to our paper entitled: Evaluation of dynamic load reduction for a tractor semi-trailer using air suspension system at all axles of semi-trailer. We have prepared a new version in order to take into account the reviewers’ remarks. The paper modifications are emphasized in yellow in the revised version. In the attachment, we provide our responses to the Reviewer's comments.
Thank you very much!

Reviewer 2 Report

In my opinion this work is very interesting, it's well written and supported by strong experimental activities.

However, in order to meet the high quality standards of MDPI publications, I suggest some further comments slight modifications.

Authors compare a mechanical leaf spring suspension with a pneumatic one, behaviour of leaf elastic elements is clearly dominated by internal friction. The effects of friction damping in this kind of suspension depends on loading conditions of the system so probably some further comments on this aspects should done.

Also it's interesting to notice that superiority of proposed pneumatic suspension is mainly related to the capability of this solution to assure also a better controlled damping, So in my opinion author should at least say something concerning the fact that also a different array of suspensions/elastic spring and dampers should produce an improvement of system response.

For what concern hysteretic phenomena due to internal friction damping i suggest to use specific operators such as the bouc wen one (you find in literature lot of works on friction modelling)

For what concern the modelling of pneumatic system, i suggest to better introduce the calculation of equivalent mechanical impedance of the suspension system by introducing some further concepts related to modelling of lumped pneumatic systems that are often better described in papers related to modelling of lumped fluidic systems like pneumatic or hydraulic brakes (here are two examples ):

1) Pugi, L., Alfatti, F., Berzi, L., Favilli, T., Pierini, M., Forrier, B., D’hondt, T., Sarrazin, M. Fast modelling and identification of hydraulic brake plants for automotive applications (2020) International Journal of Fluid Power, 21 (2), pp. 169-210. DOI: 10.13052/ijfp1439-9776.2122 2) Pugi, L., Palazzolo, A., Fioravanti, D. Simulation of railway brake plants: An application to SAADKMS freight wagons (2008) Proceedings of the Institution of Mechanical Engineers, Part F: Journal of Rail and Rapid Transit, 222 (4), pp. 321-329. DOI: 10.1243/09544097JRRT118

 In particular, linearization proposed by authors for the pneumatic system has to take count of some non-linear behaviour of pneumatic systems, like chocking that should affect their behaviour changing the values of linearized mech impedance respect to different working points.

Author Response

(The authors gave the same response as above.)
